# Metabolic syndrome, social isolation, and sarcopenia in mild cognitive impairment: A multifaceted analysis of risk factors and mediating pathways

**Shuang Deng[1], Zhongqiang Guo📙[2]\*, Min Liu[1]**

**1** School of Health, Tianhua College, Shanghai Normal University, Shanghai, China, **2** School of Nursing and Health, Henan University, Kaifeng, Henan, China

\* guozhongqiang0701@gmail.com

## Abstract

### Background

Metabolic syndrome (MetS), social isolation (SI), and sarcopenia are potential modifiable risk factors for mild cognitive impairment (MCI); however, the interactions among these factors and their mediating mechanisms remain unclear.

### Method

This study utilized data from the China Health and Retirement Longitudinal Study (CHARLS 2015), which included 2,637 subjects aged 45 and older. We employed multivariate logistic regression, a threshold effect model, bootstrap mediation analysis, and stratified analysis to investigate the associations and mechanisms between metabolic syndrome (MetS), the visceral adiposity index (VAI), the atherogenic index of plasma (AIP), nonhigh-density lipoprotein cholesterol (NHDL), social isolation (SI), accessory skeletal muscle mass (ASM), and mild cognitive impairment (MCI).

### Results

Independent effects such as social isolation (OR=1.397, 95% CI = 1.091–1.789), METS_IR (OR=0.976/unit), AIP (OR=0.593), and low ASM (OR=0.903/unit) were significantly correlated with MCI (all $P < 0.05$). The threshold effect indicates that there is a turning point for METS_IR at 27.75. The risk reduction was more pronounced when it was less than 27.752 (OR=0.905 vs. 0.982, $P = 0.002$). Mediating pathways: AIP and NHDL mediate 21.9% and 19.7% of the effects of METS_IR on MCI, respectively; social isolation mediated 3.9% of the effects of ASM (all $P < 0.05$). Population differences: The protective effect of AIP was more significant among females (OR=0.512), in rural areas (OR = 0.350), and in populations with low education (OR=0.565) ($P < 0.05$).

**Data availability statement:** All relevant data are within the paper and its Supporting information files.

**Funding:** The author(s) received no specific funding for this work.

**Competing interests:** The authors have declared that no competing interests exist.

## Conclusion

Metabolic disorders, social isolation, and sarcopenia increase the risk of MCI through independent and synergistic effects, which are partially mediated by lipid metabolism pathways. A multidimensional strategy integrating metabolic management, social support, and muscle strength intervention needs to be developed for high-risk populations such as elderly individuals and rural women.

## Introduction

Mild cognitive impairment (MCI) is considered a transitional stage between normal ageing and dementia and is characterized by impaired cognitive domains, such as memory and executive function, but basic preservation of daily living abilities [1]. With the intensification of global ageing, the early identification and intervention of MCI have become key to delaying the progression of dementia. Epidemiological data show that 10%−15% of MCI cases progress to Alzheimer's disease (AD) per year, and the cumulative conversion rate can reach 80% within 6 years [2]. This dynamic evolution highlights its importance as a precursor stage of AD.

Recent studies have revealed that metabolic syndrome (MetS), social isolation (SI), and accessory skeletal muscle mass (ASM) are three significant modifiable risk factors for mild cognitive impairment (MCI). Among them, MetS damages cerebral microvascular function through pathways such as insulin resistance, chronic inflammation, and oxidative stress, accelerating white matter hyperintensity (WMH) and neuronal degeneration [3,4].

Notably, new metabolic markers, such as the insulin resistance index (METS_IR) and the atherosclerosis index (AIP), are more sensitive than traditional indicators (such as fasting glucose) for reflecting the threshold effect of metabolic disorders on cognition. For example, an increase in the plasma triglyceride/high-density lipoprotein cholesterol (TG/HDL-C) ratio is significantly positively correlated with the rate of cognitive decline in MCI patients [5,6], but its nonlinear effects and key inflection points remain to be elucidated [7].

In addition, lipid metabolism abnormalities (such as elevated nonhigh-density lipoprotein (NHDL)) may mediate the association between MetS and MCI through changes in blood–brain barrier permeability, but this hypothesis still requires validation with multiple types of omic data [5,6].

Social isolation exacerbates cognitive decline through a dual mechanism: on the one hand, SI leads to inhibition of hippocampal neurogenesis by reducing cognitive stimuli [8]; on the other hand, SI activates the hypothalamic–pituitary–adrenal axis (HPA axis), promotes glucocorticoid release, upregulates β-secretase 1 (BACE1) expression, and accelerates β-amyloid protein (Aβ) deposition [9]. Notably, sarcopenia and SI may form a pathological cycle: decreased muscle mass leads to restricted activity, which further exacerbates the decrease in social participation, whereas SI-related neuroinflammation inhibits the secretion of myogenic neurotrophic factors (such as brain-derived neurotrophic factor (BDNF)) through the "muscle–brain axis",

damaging synaptic plasticity [10]. Animal experiments have shown that muscle-specific gene expression can directly affect the grey matter volume of the prefrontal cortex [11], but the bidirectional relationship and mediating effects between the two still need to be quantified in human studies [10,12].

Although the independent effects of the above factors are gradually becoming clear, there are still significant research gaps in terms of their synergistic effects and mechanistic pathways: 1) Nonlinear relationships: There may be a critical threshold in the dose–response curve for metabolic syndrome (MetS) indicators, such as insulin resistance (METS_IR) and the atherogenic index of plasma (AIP), in relation to the risk of mild cognitive impairment (MCI). However, existing research predominantly relies on linear assumptions, and the application of threshold recognition methods, such as restricted cubic splines, is inadequate [5,7]. 2) Mediating pathways: Does abnormal lipid metabolism (such as AIP and NHDL elevation) mediate the relationship between MetS and MCI? Does SI affect cognitive function by regulating muscle–brain axis signal transduction? This type of mechanistic hypothesis lacks longitudinal data support [9,10]. 3) Population heterogeneity: The modifying effects of urban–rural differences, education levels, and sex on risk factors have not been systematically evaluated. For example, the lack of medical resources in rural areas may lead to missed diagnoses of MCI, while women may be more susceptible to metabolic disorders because of fluctuations in oestrogen levels [13,14]. The research objectives include the following: 1) Analysing the independent/combined associations among MetS (METS_IR, VAI, AIP, and NHDL), SI, and ASM and MCI; 2) using threshold effect analysis and mediation analysis to determine the nonlinear threshold of metabolic markers and the mediating effect of social isolation; and 3) revealing the heterogeneity of risk among elderly, rural, and low-education populations through subgroup analysis. The results provide evidence-based support for multidimensional interventions for MCI and assist in the development of precise prevention and control strategies.

## Materials and methods

### Study population

The research subjects for this study are drawn from the China Health and Retirement Longitudinal Study (CHARLS), a comprehensive longitudinal study covering the whole country and targeting the population aged 45 and above. The research design and evaluation methods of CHARLS have been extensively described in previous literature [15,16]. In brief, through a multistage probability sampling method, eligible individuals were recruited from 450 communities and administrative villages across 28 provinces in China [17,18]. After enrolment, participants were required to complete a standardized questionnaire and undergo relevant physical examinations. Then, follow-up was conducted every two years. All participants provided informed consent. The consent obtained was written consent; participants signed a consent form after the interviewers thoroughly explained the purpose of the study, the procedures, potential risks and benefits, and their right to withdraw at any time. This study used publicly available data from the 2015 nationwide follow-up wave of the China Health and Retirement Longitudinal Study (CHARLS). The data were obtained through authorization from the CHARLS Data Center of Peking University on May 19, 2025. The original CHARLS study was approved by the Biomedical Ethics Review Committee of Peking University (IRB00001052−1015). The dataset used in this secondary analysis has been strictly deidentified and does not contain any personally identifying information (such as name, ID number or address). Researchers were unable to identify individual participants' identities throughout the entire process of data cleaning, analysis, and result reporting [19,20].

This study focuses mainly on the data collected by CHARLS in 2015. The initial recruitment sample size was 21095 people. During the data screening process, 5350 participants with missing dependent variable data, 12987 participants with missing independent variables, 10 participants with missing covariate alcohol consumption data, and 111 participants under 45 years old were excluded. After applying the above exclusion criteria, 2637 participants were ultimately included in the subsequent analysis (Fig 1).

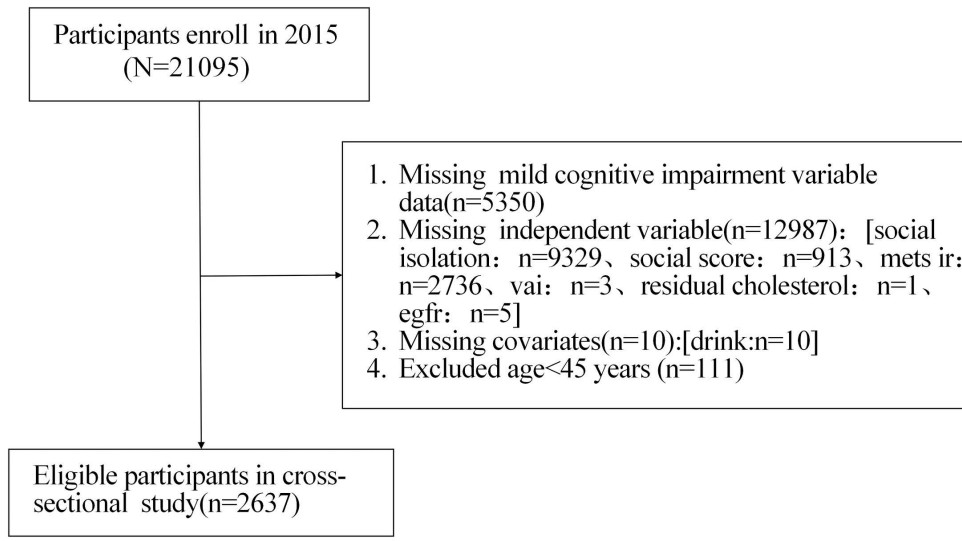

**Fig 1. Flow diagram for participant inclusion in the study.**

## Variable definition and measurement

**Outcome variable.** Mild cognitive impairment (MCI): Based on standardized cognitive assessment tools (covering dimensions such as memory, orientation, and computational ability), the Petersen criteria were used to define binary variables (0 = none, 1 = present). The specific criteria were that at least one cognitive test score must be 1.5 standard deviations below the age education norm and that daily living abilities are not impaired [21].

**Core independent variables.** Social isolation (SI): A comprehensive evaluation of social networks (such as the frequency of contact with friends and family), family structure (such as living alone), and social activity participation (such as the frequency of community activities). The total score is standardized and divided into two categories based on the median cut-off point (0 = none, 1 = present) [22].

Metabolic syndrome index:

Insulin resistance index (METS_IR): $METS\_IR = \frac{\text{Fasting blood glucose } (mg/dL) \times \text{triglyceride } (mg/dL)}{HDL–C(mg/dL)} \times \frac{BMI}{22.5}$, reflecting peripheral insulin resistance [23].

Visceral adiposity index (VAI): $VAI = \left( \frac{\text{waist circumference(cm)}}{39.68 + 1.88 \times BMI} \right) \times \left( \frac{\text{triglyceride(mmol/L)}}{1.03} \right) \times \left( \frac{1.31}{HDL–C(mmol/L)} \right)$, reflecting visceral fat accumulation [24].

Atherogenic index of plasma (AIP): $AIP = \log \left( \frac{\text{triglyceride } (mg/dL)}{HDL–C \ (mg/dL)} \right)$, used to predict cardiovascular risk.

Non-high-density lipoprotein cholesterol (NHDL): NHDL = total cholesterol–HDL-C, reflecting the total amount of atherogenic lipoproteins [25].

Accessory skeletal muscle mass (ASM): The bioelectrical impedance method (BIA) was used to determine the skeletal muscle mass (kg) of the limbs, and the Janssen formula was used to calculate it after adjusting for height, impedance value, sex, and age, as follows: $ASM = \left( \frac{\text{height}^2}{\text{electrical impedance value}} \times 0.401 \right) + (\text{gender} \times 3.825) - (\text{age} \times 0.071) + 5.102$ [26].

## Covariates

The demographic variables included age (continuous variable, with binary classification in stratified analysis as ≥ 60 years or <60 years), sex, urban vs. rural residence, and marital status.

Socioeconomic variables: Education was classified as no formal education, high school and below, and above high school.

Behavioural factors included smoking history (yes/no) and frequency of alcohol consumption (never/occasionally/frequently).

## Statistical analysis

Multivariate logistic regression: Model construction began with an unadjusted model that included only core independent variables (social isolation, metabolic indicators, and ASM).

The partially adjusted model (Adjusted I) included adjustments for age and sex.

The fully adjusted model (Adjusted II) included further adjustments for urban vs. rural residence, marital status, education level, smoking status, and alcohol consumption.

Threshold effect analysis: To explore potential nonlinear relationships between continuous metabolic indices (METS_IR, VAI, AIP, NHDL, etc.) and MCI, we conducted a two-step analysis. First, restricted cubic splines (RCSs) with likelihood ratio tests were used to visually and statistically assess the assumption of linearity for each metabolic variable. For variables where a nonlinear relationship was suggested (P for nonlinearity < 0.05), a piecewise linear regression model was fitted to identify the inflection point (K value) where the relationship between the exposure and outcome changed. The model automatically estimates the inflection point that maximizes the model likelihood. Odds ratios (ORs) and 95% confidence intervals (CIs) were then calculated separately for values on either side of the inflection point [27].

Causal mediation analysis: To test the hypothesized mediating pathways, we performed causal mediation analysis using the bootstrapping method with 1,000 bias-corrected resamples. This analysis assessed the mediating role of dyslipidaemia markers (AIP and NHDL) in the association between METS_IR (exposure) and MCI (outcome) and the mediating role of social isolation in the association between ASM (exposure) and MCI (outcome). The significance of the indirect (mediating) effect was determined by examining the 95% bootstrap confidence interval (CI). An indirect effect was considered statistically significant if this CI did not include zero. The proportion of the total effect mediated was also calculated.

Subgroup analysis and interaction tests: Stratified analyses were conducted by age (<60 vs. ≥ 60 years), sex, residence (urban vs. rural), and education level. To formally test for effect modification, interaction terms (e.g., sex*AIP) were introduced into the fully adjusted logistic regression model (Adjusted II). The statistical significance of these interaction terms was evaluated using the Wald test [28].

Statistical analysis tools included R 4.3.1 (Threshold Model and Mediation Analysis) and EmpowerStats statistical software for data cleaning and descriptive statistics. The significance level was set for bilateral tests as $\alpha = 0.05$.

## Results

### Baseline characteristics

A total of 2,637 subjects were included in the study, comprising 467 individuals in the mild cognitive impairment (MCI) group and 2,170 in the non-MCI group. Significant differences in intergroup characteristics were observed (Table 1). The metabolic indicators METS_IR (34.8 vs. 37.3), VAI (4.5 vs. 5.1), AIP (0.35 vs. 0.40), and NHDL (77.8 vs. 94.8) were lower in the MCI group than in the non-MCI group (all p < 0.05). Social isolation was significantly more prevalent in the MCI group (44.1% vs. 29.1%, p < 0.001). ASM was lower in the MCI group (17.2 vs. 18.2 kg, p < 0.001). Regarding covariates, elderly individuals (≥ 65 years: 65.7% compared to 54.0%), rural residents (73.2% versus 64.5%), and those with low education levels (43.0% vs. 31.4%) were more common in the MCI group (all p < 0.001).

### Multivariate regression analysis

In the Adjusted II model, social isolation significantly increased the risk of MCI (OR=1.397, 95% CI = 1.091–1.789, P = 0.008). For every 1-unit increase in METS_IR, the risk of MCI decreased by 2.4% (OR=0.976, 95% CI = 0.961–0.990, P = 0.001). There was a significant correlation between AIP reduction and MCI risk reduction (OR=0.593, 95%

**Table 1. Baseline characteristics of the population stratified by MCI status (n = 2637).**

| Characteristic | Without MCI (2170) | With MCI (467) | P value |
|---|---|---|---|
| **Age (years)** | | | <0.001 |
| <60 | 998 (45.991%) | 160 (34.261%) | |
| ≥60 | 1172 (54.009%) | 307 (65.739%) | |
| **Sex** | | | 0.661 |
| Male | 1209 (55.714%) | 255 (54.604%) | |
| Female | 961 (44.286%) | 212 (45.396%) | |
| **Living area** | | | <0.001 |
| Urban community | 770 (35.484%) | 125 (26.767%) | |
| Rural village | 1400 (64.516%) | 342 (73.233%) | |
| **Marital status** | | | <0.001 |
| Unmarried | 318 (14.654%) | 128 (27.409%) | |
| Married | 1852 (85.346%) | 339 (72.591%) | |
| **Education** | | | <0.001 |
| No formal education | 682 (31.429%) | 201 (43.041%) | |
| High school and below | 1457 (67.143%) | 263 (56.317%) | |
| Above high school | 31 (1.429%) | 3 (0.642%) | |
| **Smoking status** | | | 0.549 |
| No | 1088 (50.138%) | 227 (48.608%) | |
| Yes | 1082 (49.862%) | 240 (51.392%) | |
| **Alcohol consumption** | | | 0.831 |
| Low frequency | 1725 (79.493%) | 370 (79.229%) | |
| Intermediate frequency | 128 (5.899%) | 25 (5.353%) | |
| High frequency | 317 (14.608%) | 72 (15.418%) | |
| **Social isolation** | | | <0.001 |
| No | 1539 (70.922%) | 261 (55.889%) | |
| Yes | 631 (29.078%) | 206 (44.111%) | |
| **Frailty** | | | <0.001 |
| No | 1823 (84.009%) | 338 (72.377%) | |
| Yes | 347 (15.991%) | 129 (27.623%) | |
| **METS_IR** | 37.307 ± 17.188 | 34.800 ± 6.967 | 0.002 |
| **VAI** | 5.115 ± 4.395 | 4.516 ± 3.688 | 0.006 |
| **AIP** | 0.404 ± 0.283 | 0.352 ± 0.271 | <0.001 |
| **NHDL** | 94.766 ± 95.746 | 77.829 ± 81.724 | <0.001 |
| **Residual cholesterol** | 0.803 ± 0.464 | 0.753 ± 0.409 | 0.029 |
| **EGFR** | 90.850 ± 15.133 | 88.595 ± 14.727 | 0.003 |
| **Frailty index** | 4.769 ± 3.658 | 6.430 ± 4.447 | <0.001 |
| **ASM** | 18.185 ± 4.058 | 17.161 ± 4.011 | <0.001 |

CI = 0.406–0.865, $P = 0.007$). For every 1 kg decrease in ASM, the risk of MCI increased by 10.8% (OR=0.903, 95% CI = 0.865–0.944, p < 0.001) (Table 2).

## Threshold effect analysis

METS_IR showed a significant nonlinear relationship (P = 0.002), with a turning point of 27.752: below 27.752, for every 1 unit increase in METS_IR, the risk of MCI decreased by 9.5% (OR=0.905, 95% CI = 0.823–0.995, $P = 0.040$); in contrast,

**Table 2. Multiple Logistic Regression Equation.**

| Exposure | OR (95% CI) *P* value | | |
|---|---|---|---|
| | Nonadjusted | Adjusted I | Adjusted II |
| **Social isolation** | | | |
| No | 1 | 1 | 1 |
| Yes | 1.925 (1.568, 2.363) <0.001 | 1.769 (1.434, 2.183) <0.001 | 1.397 (1.091, 1.789) 0.008 |
| **METS_IR** | 0.967 (0.953, 0.982) <0.001 | 0.970 (0.955, 0.984) <0.001 | 0.976 (0.961, 0.990) 0.001 |
| **VAI** | 0.963 (0.937, 0.989) 0.006 | 0.960 (0.933, 0.988) 0.005 | 0.968 (0.941, 0.995) 0.023 |
| **AIP** | 0.513 (0.356, 0.740) <0.001 | 0.524 (0.362, 0.759) <0.001 | 0.593 (0.406, 0.865) 0.007 |
| **NHDL** | 0.998 (0.997, 0.999) <0.001 | 0.998 (0.997, 0.999) <0.001 | 0.998 (0.997, 1.000) 0.006 |
| **Residual cholesterol** | 0.762 (0.597, 0.973) 0.029 | 0.752 (0.586, 0.965) 0.025 | 0.802 (0.625, 1.030) 0.083 |
| **EGFR** | 0.991 (0.984, 0.997) 0.003 | 0.997 (0.989, 1.004) 0.363 | 0.997 (0.989, 1.004) 0.409 |
| **Frailty index** | 1.104 (1.078, 1.131) <0.001 | 1.095 (1.068, 1.123) <0.001 | 1.088 (1.061, 1.116) <0.001 |
| **Frailty** | | | |
| No | 1 | 1 | 1 |
| Yes | 2.005 (1.588, 2.531) <0.001 | 1.827 (1.436, 2.324) <0.001 | 1.718 (1.344, 2.196) <0.001 |
| **ASM** | 0.939 (0.916, 0.963) <0.001 | 0.881 (0.844, 0.919) <0.001 | 0.903 (0.865, 0.944) <0.001 |

at and above 27.752, the risk was reduced by only 1.8% (OR=0.982, 95% CI=0.966–0.998, *P*=0.028). For AIP, the inflection point was −0.038 (*P*=0.002): when the AIP was <−0.038, the MCI risk decreased by 99.8% (OR=0.002, *P*=0.001); when the AIP was ≥−0.038, the effect was not significant (OR = 0.748, p=0.156) (Table 3 and Fig 2).

## Causal mediation analysis

The mediating roles of AIP and NHDL were as follows:

AIP mediated 21.9% of the total effect of METS_IR on MCI (β=−0.0027, 95% CI=−0.0076 to −0.0001, p=0.042).

NHDL mediated effects 19.7% of the total effect of METS_IR on MCI (β=−0.0024, 95% CI=−0.0065~−0.0003, *P*=0.026).

The mediating role of social isolation was as follows:

**Table 3. Threshold effect analysis.**

| Exposure: | METS_IR | VAI | AIP | NHDL | Residual cholesterol | EGFR | Frailty index | ASM |
|---|---|---|---|---|---|---|---|---|
| Outcome: MCI | OR (95% CI) P value | OR (95% CI) P value | OR (95% CI) P value | OR (95% CI) P value | OR (95% CI) P value | OR (95% CI) P value | OR (95% CI) P value | OR (95% CI) P value |
| Model I | | | | | | | | |
| Linear effect | 0.976 (0.961, 0.990) 0.001 | 0.968 (0.941, 0.995) 0.023 | 0.593 (0.406, 0.865) 0.007 | 0.998 (0.997, 1.000) 0.006 | 0.802 (0.625, 1.030) 0.083 | 0.997 (0.989, 1.004) 0.409 | 1.088 (1.061, 1.116) <0.001 | 0.903 (0.865, 0.944) <0.001 |
| Model II | | | | | | | | |
| Folding point (K) | 27.752 | 1.238 | −0.038 | −4.924 | 0.669 | 62.814 | 0.607 | 11.630 |
| <K-segment effect 1 | 0.905 (0.823, 0.995) 0.040 | 0.212 (0.068, 0.655) 0.007 | 0.002 (0.000, 0.087) 0.001 | 0.954 (0.930, 0.979) <0.001 | 1.375 (0.526, 3.591) 0.516 | 1.035 (0.999, 1.071) 0.056 | NA | 1.054 (0.844, 1.317) 0.6418 |
| >K-segment effect 2 | 0.982 (0.966, 0.998) 0.028 | 0.973 (0.946, 1.001) 0.059 | 0.748 (0.501, 1.117) 0.156 | 0.999 (0.998, 1.000) 0.059 | 0.712 (0.511, 0.993) 0.045 | 0.990 (0.980, 0.999) 0.032 | 1.076 (1.048, 1.104) <0.001 | 0.892 (0.851, 0.935) <0.001 |
| Log likelihood ratio tests | 0.124 | 0.011 | 0.002 | <0.001 | 0.251 | 0.015 | <0.001 | 0.146 |

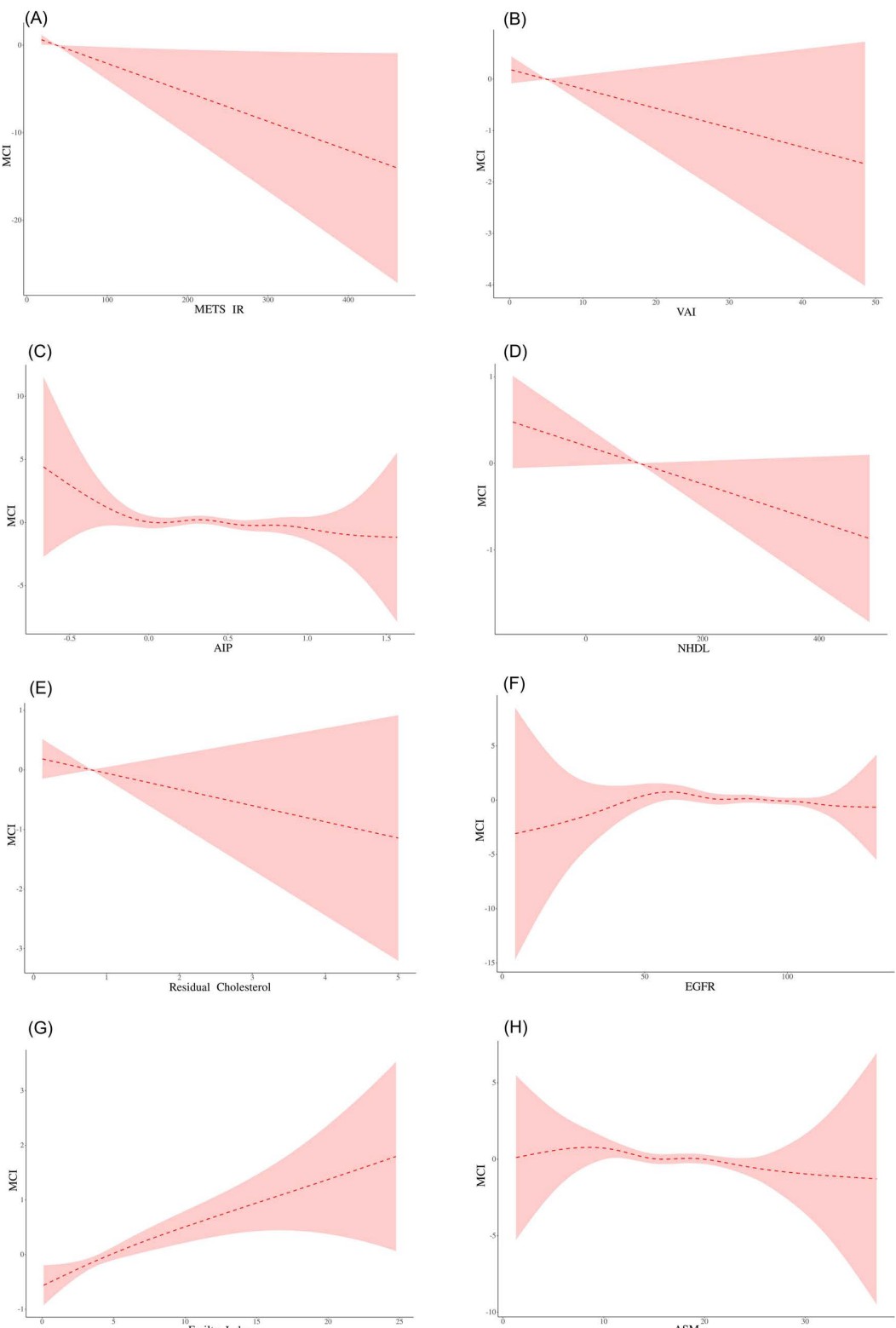

**Fig 2. Smooth curve fitting.** Smooth curve fitting relationship between (A) METS_IR and MCI, (B) VAI and MCI, (C) AIP and MCI, (D) NHDL and MCI, (E) Residual cholesterol and MCI, (F) EGFR and MCI, (G) frailty index and MCI, and (H) ASM and MCI.

ASM mediated 3.9% of the total effect of social isolation on MCI (β=−0.0034, 95% CI=−0.0072 to −0.0001, p=0.040) (Fig 3).

## Subgroup analysis

Among the population aged 65 and above, the effect of METS_IR effect was stronger (OR=0.964 vs. 0.991, p-interaction=0.002).

AIP had a more significant protective effect among females (OR=0.512 vs. males 0.713, p-interaction=0.027).

The AIP effect was more significant in rural populations (OR = 0.350) than in urban populations (0.713, p-interaction=0.004).

The protective effect of AIP was more significant in the low-education population (OR=0.565 vs. high-education 0.884, p-interaction=0.025) (Fig 4).

## Discussion

This study is based on CHARLS 2015 data and reveals the complex associations and mediating mechanisms among metabolic syndrome (MetS), social isolation, sarcopenia, and mild cognitive impairment (MCI).

The following is a multidimensional interpretation of the results based on existing literature. Regarding the complex association between metabolic syndrome and MCI, this study revealed a negative correlation between the insulin resistance index (METS_IR) and the risk of MCI (OR=0.976), which is consistent with finding that brain insulin resistance exacerbates neurodegeneration through PI3K-Akt signalling pathway disorders, leading to an increase in IRS-1 phosphorylation levels in the hippocampus, which is independently associated with Aβ plaque deposition and cognitive decline [29]. However, threshold effect analysis revealed that the protective effect was more significant when METS_IR was < 27.752 (OR=0.905), suggesting that moderate insulin resistance may maintain brain glucose metabolism through compensatory hyperinsulinaemia, whereas severe decompensation accelerates damage [27].

This nonlinear relationship is partially consistent with Reaven's "insulin resistance paradox", which suggests that compensatory hyperinsulinaemia may delay metabolic disorders, but long-term decompensation can lead to lipid toxicity and oxidative stress [30].

Mediation analysis further revealed that AIP (atherogenic index) and NHDL (non-high-density lipoprotein) mediated 21.9% and 19.7% of the effect of METS_IR, respectively (P<0.05).

These findings indicate that dyslipidaemia is the core pathway through which MetS affects cognition, complementing the mechanism proposed by De La Monte that peripheral insulin resistance transmits toxic lipids (such as sphingosine) through the liver–brain axis [31].

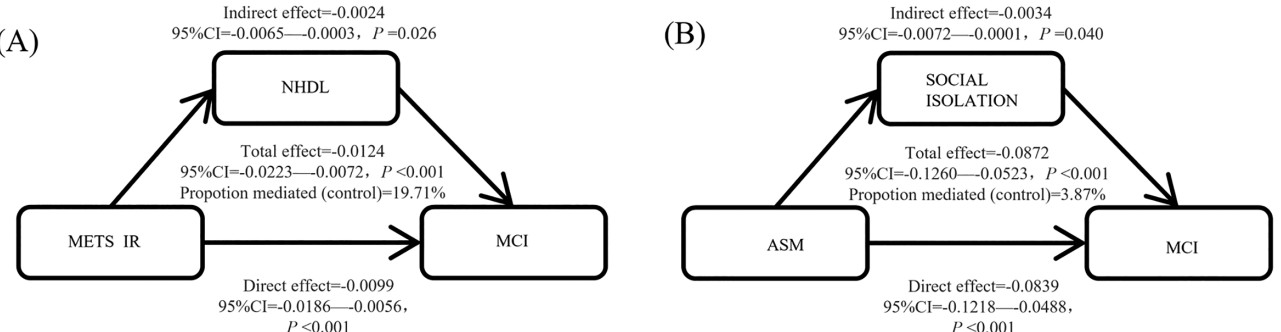

**Fig 3. Causal mediation analysis.** (A) Mediating effect analysis of NHDL as an indicator of METS_IR as it relates to MCI. (B) Mediating effect analysis of ASM as an indicator of social isolation as it relates to MCI.

| Y=MCI | | X=SOCIAL _ISOLATION | | X=METS_IR | | X=VAI | | X=AIP | | X=NHDL | |
|---|---|---|---|---|---|---|---|---|---|---|---|
| Variable | N（%） | OR 1 | P value 1 | OR 2 | P value 2 | OR 3 | P value 3 | OR 4 | P value 4 | OR 5 | P value 5 |
| AGE | | | | | | | | | | | |
| < 60 | 1158（43.9） | 1.665 (1.105, 2.509) | 0.015 | 0.991 (0.972, 1.012) | 0.402 | 0.988 (0.948, 1.029) | 0.553 | 0.884 (0.488, 1.603) | 0.685 | 0.999 (0.998, 1.001) | 0.544 |
| ≥60 | 1479（56.1） | 1.272 (0.935, 1.729) | 0.125 | 0.964 (0.945, 0.983) | < 0.001 | 0.951 (0.914, 0.989) | 0.013 | 0.448 (0.273, 0.735) | 0.002 | 0.997 (0.996, 0.999) | 0.002 |
| GENDER | | | | | | | | | | | |
| Male | 1464（55.5） | 1.339 (0.964, 1.859) | 0.081 | 0.970 (0.950, 0.990) | 0.004 | 0.948 (0.904, 0.994) | 0.028 | 0.512 (0.310, 0.847) | 0.009 | 0.998 (0.996, 1.000) | 0.015 |
| Female | 1173（44.5） | 1.496 (1.029, 2.175) | 0.035 | 0.981 (0.961, 1.002) | 0.079 | 0.981 (0.946, 1.017) | 0.291 | 0.713 (0.398, 1.276) | 0.254 | 0.999 (0.997, 1.000) | 0.153 |
| Location | | | | | | | | | | | |
| Urban Community | 895（33.9） | 1.546 (0.964, 2.479) | 0.071 | 0.962 (0.935, 0.989) | 0.006 | 0.953 (0.906, 1.003) | 0.066 | 0.350 (0.170, 0.721) | 0.004 | 0.997 (0.995, 1.000) | 0.028 |
| Rural Village | 1742（66.1） | 1.327 (0.992, 1.775) | 0.057 | 0.980 (0.963, 0.998) | 0.030 | 0.973 (0.940, 1.007) | 0.116 | 0.713 (0.456, 1.114) | 0.137 | 0.999 (0.997, 1.000) | 0.062 |
| Marital status | | | | | | | | | | | |
| Non—married | 447（17） | 1.098 (0.600, 2.009) | 0.762 | 0.967 (0.935, 1.000) | 0.052 | 0.970 (0.915, 1.027) | 0.296 | 0.608 (0.270, 1.369) | 0.230 | 0.998 (0.995, 1.001) | 0.186 |
| Married | 2190（83） | 1.482 (1.134, 1.936) | 0.004 | 0.978 (0.962, 0.994) | 0.009 | 0.967 (0.935, 0.999) | 0.044 | 0.582 (0.378, 0.896) | 0.014 | 0.998 (0.997, 1.000) | 0.016 |
| EDUCATION | | | | | | | | | | | |
| No formal education | 883（33.5） | 1.687 (1.153, 2.468) | 0.007 | 0.967 (0.943, 0.990) | 0.006 | 0.971 (0.931, 1.012) | 0.157 | 0.623 (0.342, 1.134) | 0.121 | 0.998 (0.996, 1.000) | 0.107 |
| High school and below | 1720（65.2） | 1.203 (0.862, 1.678) | 0.278 | 0.981 (0.962, 1.000) | 0.051 | 0.964 (0.927, 1.004) | 0.075 | 0.565 (0.344, 0.929) | 0.025 | 0.998 (0.997, 1.000) | 0.031 |
| Above high school | 34（1.3） | | | | | | | | | | |
| Smoking situation | | | | | | | | | | | |
| No | 1315（49.9） | 1.589 (1.116, 2.262) | 0.010 | 0.980 (0.960, 1.002) | 0.071 | 0.979 (0.944, 1.016) | 0.271 | 0.662 (0.380, 1.155) | 0.147 | 0.999 (0.997, 1.000) | 0.151 |
| Yes | 1322（50.1） | 1.233 (0.872, 1.745) | 0.236 | 0.972 (0.952, 0.993) | 0.008 | 0.950 (0.907, 0.995) | 0.031 | 0.526 (0.313, 0.887) | 0.016 | 0.998 (0.996, 0.999) | 0.013 |
| Drinking status | | | | | | | | | | | |
| Low frequency | 2095（79.4） | 1.412 (1.071, 1.863) | 0.015 | 0.977 (0.961, 0.993) | 0.005 | 0.968 (0.939, 0.999) | 0.041 | 0.597 (0.387, 0.921) | 0.020 | 0.998 (0.997, 0.999) | 0.008 |
| Intermediate frequency | 153（5.8） | 1.791 (0.573, 5.601) | 0.316 | 0.960 (0.895, 1.031) | 0.263 | 0.946 (0.819, 1.093) | 0.451 | 0.858 (0.173, 4.240) | 0.851 | 0.999 (0.994, 1.004) | 0.637 |
| High frequency | 389（14.8） | 1.198 (0.627, 2.286) | 0.585 | 0.971 (0.929, 1.014) | 0.182 | 0.983 (0.903, 1.069) | 0.686 | 0.568 (0.224, 1.441) | 0.234 | 0.999 (0.996, 1.002) | 0.716 |

| Y=MCI | | X=RESIDUALCHOLESTEROL | | X=EGFR | | X=FRAILTYINDEX | | X=FRAILTY | | X=ASM | |
|---|---|---|---|---|---|---|---|---|---|---|---|
| Variable | N（%） | OR 6 | P value 6 | OR 7 | P value 7 | OR 8 | P value 8 | OR 9 | P value 9 | OR 10 | P value 10 |
| AGE | | | | | | | | | | | |
| < 60 | 1158（43.9） | 1.018 (0.700, 1.480) | 0.925 | 1.004 (0.989, 1.020) | 0.579 | 1.125 (1.073, 1.180) | < 0.001 | 2.178 (1.371, 3.461) | 0.001 | 0.924 (0.859, 0.994) | 0.034 |
| ≥60 | 1479（56.1） | 0.673 (0.478, 0.945) | 0.022 | 0.995 (0.986, 1.003) | 0.236 | 1.072 (1.040, 1.105) | < 0.001 | 1.566 (1.173, 2.091) | 0.002 | 0.890 (0.843, 0.940) | < 0.001 |
| GENDER | | | | | | | | | | | |
| Male | 1464（55.5） | 0.793 (0.558, 1.127) | 0.196 | 0.998 (0.988, 1.008) | 0.654 | 1.106 (1.067, 1.146) | < 0.001 | 1.998 (1.411, 2.829) | < 0.001 | 0.899 (0.849, 0.952) | < 0.001 |
| Female | 1173（44.5） | 0.804 (0.561, 1.152) | 0.234 | 0.996 (0.984, 1.007) | 0.470 | 1.072 (1.033, 1.112) | < 0.001 | 1.502 (1.060, 2.128) | 0.022 | 0.904 (0.845, 0.968) | 0.004 |
| Location | | | | | | | | | | | |
| Urban Community | 895（33.9） | 0.699 (0.442, 1.104) | 0.124 | 1.012 (0.998, 1.026) | 0.094 | 1.068 (1.015, 1.123) | 0.011 | 1.611 (1.001, 2.591) | 0.050 | 0.882 (0.814, 0.956) | 0.002 |
| Rural Village | 1742（66.1） | 0.854 (0.634, 1.151) | 0.300 | 0.989 (0.980, 0.998) | 0.021 | 1.094 (1.062, 1.127) | < 0.001 | 1.739 (1.304, 2.319) | < 0.001 | 0.910 (0.864, 0.958) | < 0.001 |
| Marital status | | | | | | | | | | | |
| Non—married | 447（17） | 1.048 (0.633, 1.736) | 0.855 | 0.993 (0.979, 1.008) | 0.390 | 1.070 (1.018, 1.124) | 0.007 | 1.267 (0.789, 2.037) | 0.327 | 0.888 (0.811, 0.973) | 0.011 |
| Married | 2190（83） | 0.735 (0.546, 0.989) | 0.042 | 0.998 (0.989, 1.007) | 0.688 | 1.095 (1.063, 1.128) | < 0.001 | 1.904 (1.431, 2.535) | < 0.001 | 0.909 (0.865, 0.955) | < 0.001 |
| EDUCATION | | | | | | | | | | | |
| No formal education | 883（33.5） | 0.792 (0.531, 1.181) | 0.253 | 0.988 (0.977, 0.999) | 0.030 | 1.087 (1.047, 1.128) | < 0.001 | 1.531 (1.072, 2.187) | 0.019 | 0.865 (0.806, 0.930) | < 0.001 |
| High school and below | 1720（65.2） | 0.824 (0.595, 1.141) | 0.243 | 1.004 (0.993, 1.014) | 0.479 | 1.090 (1.052, 1.130) | < 0.001 | 1.896 (1.349, 2.666) | < 0.001 | 0.926 (0.875, 0.980) | 0.007 |
| Above high school | 34（1.3） | | | 1.082 (0.844, 1.387) | 0.534 | 0.883 (0.220, 3.540) | 0.861 | | | 0.514 (0.099, 2.654) | 0.427 |
| Smoking situation | | | | | | | | | | | |
| No | 1315（49.9） | 0.835 (0.591, 1.178) | 0.304 | 0.998 (0.987, 1.009) | 0.771 | 1.067 (1.030, 1.106) | 0.000 | 1.516 (1.072, 2.142) | 0.019 | 0.910 (0.854, 0.970) | 0.004 |
| Yes | 1322（50.1） | 0.762 (0.526, 1.104) | 0.150 | 0.996 (0.986, 1.006) | 0.441 | 1.109 (1.069, 1.151) | < 0.001 | 1.922 (1.352, 2.733) | < 0.001 | 0.899 (0.847, 0.954) | < 0.001 |
| Drinking status | | | | | | | | | | | |
| Low frequency | 2095（79.4） | 0.817 (0.620, 1.078) | 0.153 | 0.995 (0.987, 1.003) | 0.250 | 1.087 (1.057, 1.117) | < 0.001 | 1.596 (1.221, 2.087) | < 0.001 | 0.919 (0.875, 0.964) | < 0.001 |
| Intermediate frequency | 153（5.8） | 0.303 (0.057, 1.603) | 0.160 | 0.997 (0.963, 1.032) | 0.879 | 1.238 (1.076, 1.424) | 0.003 | 5.898 (1.788, 19.456) | 0.004 | 0.789 (0.633, 0.983) | 0.035 |
| High frequency | 389（14.8） | 0.919 (0.478, 1.767) | 0.800 | 1.003 (0.983, 1.025) | 0.748 | 1.048 (0.961, 1.143) | 0.292 | 1.989 (0.919, 4.302) | 0.081 | 0.839 (0.743, 0.948) | 0.005 |

**Fig 4. Subgroup analysis.**

In addition, the protective effects of AIP and NHDL were more significant in the elderly population (>65 years old) (OR=0.448–0.673), supporting the hypothesis that the brain becomes more sensitive to metabolic disorders with age [29].

Regarding the independent roles of social isolation and sex/urban–rural differences, social isolation increased the risk of MCI by 39.7% (adjusted II model), which is consistent with the findings of the Iranian TLGS study showing that during the COVID-19 pandemic, the mental health scores of diabetes patients decreased by 5 points because of social isolation, and the incidence of depression symptoms reached 60.8% [32].

However, this study revealed that the impact of social isolation was more significant in the married population (OR=1.482) and the low-education population (OR=1.687), which may be due to changes in marital status (such as widowhood) or insufficient educational resources exacerbating psychological stress.

This is in contrast to the "family support buffering effect" studied in Tehran, which shows that while family support can improve women's physical health indicators (such as blood pressure), it cannot offset the negative effects of social deprivation on mental health [33].

For example, among female housewives in Tehran, there is a significant correlation between family support and mental health scores (Kendall's tau = 0.321) [34], but the results of this study suggest that psychological stress may exacerbate neuroinflammation through activation of the HPA axis, independent of family support.

In terms of urban–rural differences, rural residents have a greater risk of social isolation (OR=1.327), reflecting the combined effects of rural social resource scarcity and weak health awareness.

This phenomenon is consistent with CHARLS research: the social activity participation of elderly people in rural China is relatively low and positively correlated with the risk of functional disability (HR = 1.15) [35].

Sarcopenia plays a dual role, mediating direct injury and social isolation. ASM reduction directly increases the risk of MCI (OR=0.903), which is consistent with the underlying pathological mechanisms of mitochondrial dysfunction and oxidative stress caused by rhabdomyolysis [36].

In addition, this study revealed for the first time that sarcopenia indirectly affects MCI through social isolation (mediation ratio, 3.9%; $P$ = 0.040).

The decline in muscle function limits social activity participation, forming a vicious cycle of "physical function limitation–social isolation–cognitive decline".

This mechanism is consistent with the observation in the TLGS study that gait disorders diminish the diversity of social participation (HR = 0.78) [35,37]. However, this study specifically quantified the mediating effect within the East Asian population, providing a new target for intervention strategies.

The stratified results have public health implications. The protective effect of metabolic indicators, such as AIP and NHDL, was more significant among those older than 65 years, which may be related to a decrease in brain insulin receptor density and an increase in blood–brain barrier permeability in elderly individuals.

AIP had a stronger protective effect in females (OR=0.512 vs. 0.713 in males).

The risk of sarcopenia was lower in the more highly educated population (OR=0.926 vs. 0.865 in the low-educated population), indicating that education promotes muscle maintenance behaviour through health literacy (such as resistance training) [35].

Limitations of this study include the cross-sectional design, which prevents the establishment of causality; however, the Tehran study confirmed the lagged damage of social isolation on mental health through longitudinal data [33].

The assessment of sarcopenia relies on bioelectrical impedance and does not include muscle function indicators (such as grip strength), which may underestimate its impact. Although the measurement of limb skeletal muscle mass (ASM) using bioelectrical impedance in this study can effectively reflect muscle mass [38], the lack of inclusion of muscle function indicators (such as grip strength and walking speed) is a major limitation. According to the consensus of the European Working Group on sarcopenia in the elderly (EWGSOP2) [39], the diagnosis of sarcopenia requires both a decrease in muscle mass and a decrease in muscle function/performance. Owing to the lack of systematic collection of

muscle function data in the CHARLS 2015 database, future research should integrate tools such as grip strength tests or the Short Physical Performance Battery (SPPB) to more comprehensively evaluate the impact of muscle loss on cognition and further validate the pathway of "limited physical function → social isolation → cognitive decline" proposed in this study.

Residual confounding factors, such as uncontrolled intake of omega-3 fatty acids, which reduce the levels of inflammatory factors (such as IL-6) by inhibiting the NF-κB pathway, may interfere with the association between metabolic indicators and MCI [40,41].

Regarding future research, multimodal neuroimaging (such as fMRI–PET fusion technology) can be used to analyse the "muscle metabolism–brain" axis, as previous studies have shown that multimodal data can improve the accuracy of AD diagnosis by 10% [42].

Additionally, randomized trials can be designed to verify the preventive effects of a combined intervention (such as social activity+resistance training) on MCI, referring to the "instrumental support enhances social participation" strategy proposed by the CLSA study [43].

## Conclusions

This study utilizes data from the Chinese CHARLS cohort to systematically investigate the combined effects and mediating mechanisms of metabolic syndrome (MetS), social isolation, and sarcopenia on mild cognitive impairment (MCI). The main findings are as follows: the independent and interactive effects of three risk factors, metabolic disorders (decreased METS_IR, decreased AIP, and decreased NHDL), were significantly associated with MCI risk, with AIP and NHDL mediating 21.9% and 19.7% of metabolic damage effects (P<0.05), respectively, confirming the theorized "lipid metabolism–brain health axis" [44].

Social isolation increased the risk of MCI by 39.7% (OR=1.397), and the effect was stronger among married and less educated populations, challenging the traditional "marriage protection hypothesis" [45].

Sarcopenia (ASM reduction) not only directly increased the risk of MCI (OR=0.903) but also indirectly contributed to 3.9% of the effect through social isolation (P=0.040), quantifying for the first time the pathway through which decreased muscle function impairs cognition by limiting social activity.

Targeted interventions can be guided by population heterogeneity. In elderly populations (>60 years old), the protective effects of metabolic indicators (AIP and NHDL) were more significant (OR=0.448–0.673), and priority should be given to controlling lipid disorders in these populations.

AIP had a stronger protective effect on cognition in females (OR=0.512 vs. 0.713 for males), supporting the hypothesis of an oestrogen–lipid interaction mechanism [46].

The risk of social isolation was high among those with a low education level (OR=1.687), and the effect of sarcopenia was high (ASM OR=0.865), indicating the need to integrate health literacy education with community support.

This study verified the role of several factors in the occurrence of MCI (as shown in Fig 3), providing a basis for the following intervention strategies: Regarding metabolic management, for MetS patients, controlling the atherogenic index (AI P<0. 038) may be more effective than simply reducing glucose to prevent cognitive decline [47].

Designing a "social exercise joint intervention" (such as group resistance training+community activities) for those with sarcopenia and in elderly populations could simultaneously improve physical function and promote social participation.

Finally, to support precision public health, priority should be given to projects strengthening cognitive screening and social connectivity for rural and low-income populations to address health inequalities.

## Supporting information

**S1 Data.  Raw data.** The raw data mentioned in this article are all located in this file.
(XLSX)

**S1 File. README.** The data interpretation of the raw data mentioned in this article is located in this file.
(DOCX)

## Generative AI statement

The author(s) declare that no generative AI was used in the creation of this manuscript.

## Publishers' notes

All claims expressed in this article are solely those of the authors and do not necessarily represent those of their affiliated organizations or those of the publisher, the editors and the reviewers. Any product that may be evaluated in this article or claim that may be made by its manufacturer is not guaranteed or endorsed by the publisher.

## Author contributions

**Conceptualization:** Zhongqiang Guo.

**Data curation:** Shuang Deng, Zhongqiang Guo.

**Formal analysis:** Shuang Deng, Zhongqiang Guo.

**Investigation:** Shuang Deng, Zhongqiang Guo.

**Methodology:** Shuang Deng.

**Project administration:** Min Liu.

**Software:** Zhongqiang Guo, Min Liu.

**Supervision:** Min Liu.

**Validation:** Zhongqiang Guo, Min Liu.

**Visualization:** Zhongqiang Guo, Min Liu.

**Writing – original draft:** Shuang Deng, Zhongqiang Guo.

**Writing – review & editing:** Zhongqiang Guo.

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
