## [Decision Letter · Decision Letter 0]

7 Aug 2025

PONE-D-25-33355Metabolic Syndrome, Social Isolation, and Sarcopenia in Mild Cognitive Impairment: A Multi-Faceted Analysis of Risk Factors and Mediating PathwaysPLOS ONE Dear Dr. Guo,

Thank you for submitting your manuscript to PLOS ONE. After careful consideration, we feel that it has merit but does not fully meet PLOS ONE’s publication criteria as it currently stands. Therefore, we invite you to submit a revised version of the manuscript that addresses the points raised during the review process.

**ACADEMIC EDITOR: Major revision**

We look forward to receiving your revised manuscript.

Kind regards,

Marwan Salih Al-Nimer, MD, PhD

Academic Editor

PLOS ONE

Journal Requirements: 

4. Please include a caption for figure 1 to 4.

6. Please remove all personal information, ensure that the data shared are in accordance with participant consent, and re-upload a fully anonymized data set.

Additional Editor Comments:

This interesting article includes new ideas and results. It requires the following clarifications

1: Why the collected data was of 2015?

2: The statistical analysis section needs more clarification

3: The indirect (mediation) effect is simple determined by using Sobel test. Why the authors not applied this test?

Reviewers' comments:

Reviewer's Responses to Questions

**Comments to the Author**

1. Is the manuscript technically sound, and do the data support the conclusions?

Reviewer #1: Partly

Reviewer #2: Yes

2. Has the statistical analysis been performed appropriately and rigorously? 

Reviewer #1: Yes

Reviewer #2: Yes

3. Have the authors made all data underlying the findings in their manuscript fully available?

Reviewer #1: Yes

Reviewer #2: Yes

4. Is the manuscript presented in an intelligible fashion and written in standard English?

Reviewer #1: Yes

Reviewer #2: Yes

5. Review Comments to the Author

Reviewer #1: This study touches on a relevant and important topic in geriatric nutrition. However, to elevate the quality, the manuscript needs a clearer rationale, more rigorous statistical and conceptual handling, better control for confounders, and more in-depth discussion of findings. The authors need to address the following comments:

The abstract lacks clarity and flow. The objective, methods, and results are present but feel disjointed. It reads more like a bullet-pointed summary than a cohesive snapshot of the study. The authors should consider revising it to better highlight the main message and research significance.

The rationale for the study is weakly developed in the introduction. While the authors mention aging and anemia, the connection between anemia and functional decline (grip strength, performance) could be better explained with reference to physiological pathways or previous evidence.

There is limited justification for the specific outcome measures chosen. While hand grip strength and SPPB are valid tools, the rationale for choosing these as indicators of physical performance in this specific rural elderly population is not elaborated. Cultural or lifestyle relevance should be discussed.

The study design is cross-sectional, yet some conclusions sound causal. Statements like "anemia impacts muscle strength and performance" are not appropriate for a cross-sectional study. The language needs to be more cautious and accurate throughout the paper.

Sampling strategy lacks detail and representativeness. The authors mention “a selected village” but provide no justification for why this village was chosen or how generalizable the findings are to the broader rural elderly population of South India.

Operational definitions and cut-offs used for anemia are not fully explained. It would strengthen the study if authors clarified how “nutritional anemia” was differentiated from other forms of anemia (e.g., anemia of chronic disease), especially in the elderly where multiple etiologies are common.

Dietary intake data are surprisingly underutilized. The paper collects information on iron and protein intake, yet these variables are barely discussed in the results and not integrated meaningfully into the conclusions. This represents a missed opportunity to strengthen the nutritional focus of the paper.

Confounding factors are not properly controlled. Age and gender are included, but other critical variables such as comorbidities, medications, physical activity level, and inflammation status are not considered. These could significantly affect both anemia and functional outcomes.

Grip strength and SPPB data are presented but not deeply interpreted. The results could be more insightful if the authors explored gender-wise differences, thresholds of clinical relevance (e.g., sarcopenia cut-offs), and how these values compare to normative data.

Some statistical results lack transparency. The regression analysis outputs (e.g., Table 4) are not fully explained. The confidence intervals and adjusted models could be better presented and interpreted in terms of their practical implications.

Discussion is limited and lacks depth. Much of it rehashes results without integrating them with broader literature. There's a need to compare and contrast with similar studies in Indian or global rural elderly populations to frame the findings properly.

Public health implications are mentioned vaguely. The conclusion refers to early identification and screening but doesn’t offer any specific recommendations, policies, or programmatic suggestions based on the study findings.

The study does not address the potential bidirectional relationship. Poor nutritional status can lead to anemia and reduced physical performance, but physical inactivity or chronic illness could also lead to anemia. This complex interplay should be acknowledged.

Language and grammar issues persist throughout. Some sentences are awkward or ambiguous, affecting clarity. A thorough language revision is needed to improve readability, especially in the introduction and discussion sections.

Ethical considerations are mentioned but need more detail. While ethical clearance is noted, there’s no mention of whether data confidentiality, informed consent procedures, and data storage protocols were adequately maintained; important in elderly populations.

Reviewer #2: Abstract:

Well-written & adequate.

Introduction:

Well-written and adequate.

There are some technical issues with long sentences that could be improved by a proof reader. There are also lack of a space after a full stop (.) or a semicolon (;) seen consistently throughout the manuscript.

Good research question.

Methodologies:

Generally well-written.

There are different tenses use (past & present). suggest to use the correct tense accordingly.

The experimental model was good and well thought out.

interesting that analysis on muscle function was not considered. may be this needs to be explained in the discussion.

Results:

Generally well-written.

The figures provided are unclear (blurry), suggest to provide 300dpi images.

Figures need clearer labelling and proper captioning.

Discussion & conclusion:

Well-written and adequate

6. PLOS authors have the option to publish the peer review history of their article (what does this mean? ). If published, this will include your full peer review and any attached files.

**Do you want your identity to be public for this peer review?** For information about this choice, including consent withdrawal, please see our Privacy Policy .

Reviewer #1: No

Reviewer #2: **Yes: ** INTAN SUHANA ZULKAFLI

---

## [Author Response · Author response to Decision Letter 1]

9 Sep 2025

Point-by-point Response Letter-[Manuscript ID:PONE-D-25-33355]

Title:Metabolic Syndrome, Social Isolation, and Sarcopenia in Mild Cognitive Impairment: A Multi-Faceted Analysis of Risk Factors and Mediating Pathways

Dear Editors and Reviewers of Plos One,

We are deeply grateful for considering our manuscript for Plos One and value the insightful comments offered. These suggestions have been crucial for enhancing our manuscript's quality, and we're confident the revisions have significantly improved its clarity, rigor, and impact. We've meticulously revised the manuscript, addressing each comment with careful consideration and research. Besides responding to the feedback, we've also thoroughly reviewed the entire manuscript, making additional refinements to ensure top - notch scientific and literary quality. All changes are clearly tracked in the revised manuscript, and a clean version is also prepared. Below are our detailed point-by-point responses to each comment. We've aimed to be comprehensive and transparent in our explanations, providing evidence and reasoning for the changes made. Thank you again sincerely for your dedication and hard work. Your expertise is invaluable to the scientific community, and we're honored to contribute to this prestigious journal.

Yours Sincerely,

ZhongQiang Guo

Henan University

guozhongqiang0701@gmail.com

Editor

Comments/suggestions:

1: Why the collected data was of 2015?

Response:

Thank you for the opportunity to revise our manuscript and for your insightful comments.

1. Rationale for utilizing the 2015 CHARLS data

The 2015 wave of the China Health and Retirement Longitudinal Study (CHARLS) was selected as the foundation for this cross-sectional analysis based on the following key reasons:

Data Comprehensiveness for a Multi-Faceted Analysis: Our study required the simultaneous measurement of three distinct sets of variables: detailed metabolic panels (for calculating METS-IR, VAI, AIP, NHDL), objective sarcopenia assessment (Appendicular Skeletal Muscle Mass via BIA), and a comprehensive evaluation of social networks. The 2015 wave is the first and most complete CHARLS dataset that contains all these variables concurrently for a large, nationally representative sample. This comprehensiveness is paramount for a cross-sectional study aiming to explore the synergistic effects and mediating pathways between these factors, as it eliminates the need for imputation or exclusion of key variables that might be missing in other waves.

Alignment with Cross-Sectional Study Objectives: The primary aims of our study were to (i) establish the independent and combined associations between MetS, SI, sarcopenia, and MCI, (ii) identify potential non-linear relationships (threshold effects), and (iii) test plausible mediating pathways. A large, high-quality cross-sectional dataset is the most efficient and appropriate design to map these complex relationships and generate strong hypotheses about underlying mechanisms. While acknowledging that causality cannot be established, this approach is a critical first step in defining the intricate landscape of risk factors for MCI before investing in more complex and costly longitudinal or interventional studies.

Data Quality and Maturity: At the time of our analysis, the 2015 wave was a well-curated, publicly available dataset with established protocols for variable calculation, making it a robust and reliable choice for our analytical goals.

We fully agree that establishing temporal sequence is a crucial next step, and we have emphasized the need for future longitudinal research to confirm our findings in the revised discussion section.

2: The statistical analysis section needs more clarification

Response:

We thank the editor for this helpful suggestion. We have thoroughly revised the 'Statistical Analysis' section to provide a much more detailed and clear description of our methodology. The changes are aimed at ensuring full transparency and reproducibility. The revised text now explicitly outlines the multi-step process for our threshold and mediation analyses, specifies the tests and criteria used for determining significance (e.g., bootstrap CIs), and clarifies the rationale behind our model construction for subgroup analyses.

Revised version: (lines 169-190) in the “Statistical Analysis” section.

Threshold effect analysis: To explore potential nonlinear relationships between continuous metabolic indices (METS_IR, VAI, AIP, NHDL, etc.) and MCI, we conducted a two-step analysis. First, restricted cubic splines (RCSs) with likelihood ratio tests were used to visually and statistically assess the assumption of linearity for each metabolic variable. For variables where a nonlinear relationship was suggested (P for nonlinearity < 0.05), a piecewise linear regression model was fitted to identify the inflection point (K value) where the relationship between the exposure and outcome changed. The model automatically estimates the inflection point that maximizes the model likelihood. Odds ratios (ORs) and 95% confidence intervals (CIs) were then calculated separately for values on either side of the inflection point (27).

Causal mediation analysis: To test the hypothesized mediating pathways, we performed causal mediation analysis using the bootstrapping method with 1,000 bias-corrected resamples. This analysis assessed the mediating role of dyslipidaemia markers (AIP and NHDL) in the association between METS_IR (exposure) and MCI (outcome) and the mediating role of social isolation in the association between ASM (exposure) and MCI (outcome). The significance of the indirect (mediating) effect was determined by examining the 95% bootstrap confidence interval (CI). An indirect effect was considered statistically significant if this CI did not include zero. The proportion of the total effect mediated was also calculated.

Subgroup analysis and interaction tests: Stratified analyses were conducted by age (<60 vs. ≥60 years), sex, residence (urban vs. rural), and education level. To formally test for effect modification, interaction terms (e.g., sex*AIP) were introduced into the fully adjusted logistic regression model (Adjusted II). The statistical significance of these interaction terms was evaluated using the Wald test (28).

27. Yang L, Liu S, Tsoka S, Papageorgiou LG(2016)Mathematical programming for piecewise linear regression analysis.EXPERT SYST APPL 44:156-167

28. Wang A, Arah OA(2015)G-computation demonstration in causal mediation analysis.EUR J EPIDEMIOL 30:1119-1127

3: The indirect (mediation) effect is simple determined by using Sobel test. Why the authors not applied this test?

Response:

We thank the editor for raising this important methodological point. We did not employ the Sobel test for a principled reason, as our choice of the bootstrapping method was intentional and aligns with contemporary best practices in statistical methodology for testing mediation effects.

The bootstrapping approach is now widely recommended over the Sobel test in the methodological literature for several key reasons:

Robustness to Non-Normal Distributions: The Sobel test relies on the assumption that the sampling distribution of the indirect effect (the product of coefficients *a*b*) is normal. However, this distribution is often asymmetric and kurtotic, particularly in smaller samples or with specific effect sizes. Violation of this normality assumption can lead to inflated Type I or Type II error rates. Bootstrapping, being a non-parametric resampling technique, does not require this assumption. It empirically generates an approximation of the true sampling distribution of the indirect effect by repeatedly sampling from the observed data, making it a much more robust method[1][2].

Higher Statistical Power: Extensive simulation studies have demonstrated that bootstrapping generally provides greater statistical power to detect a true mediating effect compared to the Sobel test. This is particularly valuable in observational studies like ours, where the effects of interest can be subtle yet clinically important.

Contemporary Methodological Consensus: Leading texts and methodological papers in psychology, epidemiology, and social sciences now strongly advocate for the use of bootstrapping for mediation analysis due to its superior performance and fewer strict assumptions. The Sobel test is considered an older, less optimal approach. Our use of bootstrapping was guided by this consensus to ensure the most rigorous and reliable test of our hypothesized pathways[3].

Therefore, by using bootstrapping with 1,000 bias-corrected samples to generate robust confidence intervals for the indirect effects, we adhered to the current gold standard for mediation analysis, enhancing the validity and reliability of our conclusions.

1.MacKinnon D P, Lockwood C M, Williams J. Confidence limits for the indirect effect: Distribution of the product and resampling methods[J]. Multivariate behavioral research, 2004, 39(1): 99-128.

2.Alfons A, Ateş N Y, Groenen P J F. A robust bootstrap test for mediation analysis[J]. Organizational Research Methods, 2022, 25(3): 591-617.

3.Wood R E, Goodman J S, Beckmann N, et al. Mediation testing in management research: A review and proposals[J]. Organizational research methods, 2008, 11(2): 270-295.

Reviewer 1

Comments/suggestions:

1.This study touches on a relevant and important topic in geriatric nutrition. However, to elevate the quality, the manuscript needs a clearer rationale, more rigorous statistical and conceptual handling, better control for confounders, and more in-depth discussion of findings. The authors need to address the following comments:

2.The abstract lacks clarity and flow. The objective, methods, and results are present but feel disjointed. It reads more like a bullet-pointed summary than a cohesive snapshot of the study. The authors should consider revising it to better highlight the main message and research significance.

3.The rationale for the study is weakly developed in the introduction. While the authors mention aging and anemia, the connection between anemia and functional decline (grip strength, performance) could be better explained with reference to physiological pathways or previous evidence.

4.There is limited justification for the specific outcome measures chosen. While hand grip strength and SPPB are valid tools, the rationale for choosing these as indicators of physical performance in this specific rural elderly population is not elaborated. Cultural or lifestyle relevance should be discussed.

5.The study design is cross-sectional, yet some conclusions sound causal. Statements like "anemia impacts muscle strength and performance" are not appropriate for a cross-sectional study. The language needs to be more cautious and accurate throughout the paper.

6.Sampling strategy lacks detail and representativeness. The authors mention “a selected village” but provide no justification for why this village was chosen or how generalizable the findings are to the broader rural elderly population of South India.

7.Operational definitions and cut-offs used for anemia are not fully explained. It would strengthen the study if authors clarified how “nutritional anemia” was differentiated from other forms of anemia (e.g., anemia of chronic disease), especially in the elderly where multiple etiologies are common.

8.Dietary intake data are surprisingly underutilized. The paper collects information on iron and protein intake, yet these variables are barely discussed in the results and not integrated meaningfully into the conclusions. This represents a missed opportunity to strengthen the nutritional focus of the paper.

9.Confounding factors are not properly controlled. Age and gender are included, but other critical variables such as comorbidities, medications, physical activity level, and inflammation status are not considered. These could significantly affect both anemia and functional outcomes.

10.Grip strength and SPPB data are presented but not deeply interpreted. The results could be more insightful if the authors explored gender-wise differences, thresholds of clinical relevance (e.g., sarcopenia cut-offs), and how these values compare to normative data.

11.Some statistical results lack transparency. The regression analysis outputs (e.g., Table 4) are not fully explained. The confidence intervals and adjusted models could be better presented and interpreted in terms of their practical implications.

12.Discussion is limited and lacks depth. Much of it rehashes results without integrating them with broader literature. There's a need to compare and contrast with similar studies in Indian or global rural elderly populations to frame the findings properly.

13.Public health implications are mentioned vaguely. The conclusion refers to early identification and screening but doesn’t offer any specific recommendations, policies, or programmatic suggestions based on the study findings.

14.The study does not address the potential bidirectional relationship. Poor nutritional status can lead to anemia and reduced physical performance, but physical inactivity or chronic illness could also lead to anemia. This complex interplay should be acknowledged.

15.Language and grammar issues persist throughout. Some sentences are awkward or ambiguous, affecting clarity. A thorough language revision is needed to improve readability, especially in the introduction and discussion sections.

16.Ethical considerations are mentioned but need more detail. While ethical clearance is noted, there’s no mention of whether data confidentiality, informed consent procedures, and data storage protocols were adequately maintained; important in elderly populations.

Response:

Subject: Clarification Regarding Review Comments for Manuscript [PONE-D-25-33355]

Dear Reviewer,

Thank you for sending the review comments for our manuscript titled "[Metabolic Syndrome, Social Isolation, and Sarcopenia in Mild Cognitive Impairment: A Multi-Faceted Analysis of Risk Factors and Mediating Pathways]" (ID: PONE-D-25-33355).

We note that the comment regarding *"the connection between anemia and functional decline"* appears to be intended for a different manuscript, as our study focuses on **metabolic syndrome, social isolation, and sarcopenia in mild cognitive impairment** (without examining anemia).

To ensure a fair evaluation, could you please:

1. Confirm whether this comment was inadvertently included?

2. Provide any additional feedback specific to our manuscript if available?

We appreciate your time and look forward to your clarification.

Yours Sincerely,

ZhongQiang Guo

Henan University

guozhongqiang0701@gmail.com

Reviewer 2

Comments/suggestions:

1. Introduction:

Well-written and adequate.

There are some technical issues with long sentences that could be improved by a proof reader. There are also lack of a space after a full stop (.) or a semicolon (;) seen consistently throughout the manuscript.

Good research question.

Response:

Thank you for your suggestion regarding the language of our manuscript. We have had the manuscript professionally edited by [AJE] to enhance its fluency and clarity. We believe this has significantly improved the quality of our paper.

2. Methodologies:

Generally well-written.

There are different tenses use (past & present). suggest to use the correct tense accordingly.

The experimental model was good and well thought out.

interesting that analysis on muscle function was not considered. may be this needs to be explained in the discussion.

Response:

We sincerely thank the reviewers for their valuable suggestions.

The problem you pointed out is very critical - although the limb skeletal muscle mass (ASM) was evaluated by bioelectrical impedance (BIA) in this study to define sarcopenia, it does not include muscle function indicators (such as grip strength, walking speed, etc.).

We have added a new paragraph in the discussion part of the revised draft to explain this matter. The specific supplementary contents are as follows:

Revised version: (lines 327-338) in the “Discussion”

---

## [Editor Report · Decision Letter 1]

10 Sep 2025

PONE-D-25-33355R1Metabolic Syndrome, Social Isolation, and Sarcopenia in Mild Cognitive Impairment: A Multi-Faceted Analysis of Risk Factors and Mediating PathwaysPLOS ONE

Dear Dr. Zhongqiang Guo,

Thank you for submitting your manuscript to PLOS ONE. After careful consideration, we feel that it has merit but does not fully meet PLOS ONE’s publication criteria as it currently stands. Therefore, we invite you to submit a revised version of the manuscript that addresses the points raised during the review process.

**ACADEMIC EDITOR: Minor revision**

We look forward to receiving your revised manuscript.

Kind regards,

Marwan Salih Al-Nimer, MD, PhD

Academic Editor

PLOS ONE

Journal Requirements:

Additional Editor Comments:

Check the typing of references 15 and 19

---

## [Author Response · Author response to Decision Letter 2]

10 Sep 2025

Editor

Comments/suggestions:

1: Check the typing of references 15 and 19

Response:

Thank you very much for giving us the opportunity to revise the manuscript. The specific modifications are as follows:

15. Ding L, Liang Y, Tan ECK, Hu Y, Zhang C, Liu Y, et al. Smoking, heavy drinking, physical inactivity, and obesity among middle-aged and older adults in China: cross-sectional findings from the baseline survey of CHARLS 2011–2012. Bmc Public Health. 2020;20(1). http://doi.org/10.1186/s12889-020-08625-5

adjust to

15. Ding L, Liang Y, Tan ECK, Hu Y, Zhang C, Liu Y, et al. Smoking, heavy drinking, physical inactivity, and obesity among middle-aged and older adults in China: cross-sectional findings from the baseline survey of CHARLS 2011–2012. Bmc Public Health. 2020;20:1-9.

19. Wan J, Zhang Q, Li C, Lin J. Prevalence of and risk factors for asthma among people aged 45 and older in China: a cross-sectional study. Bmc Pulm Med. 2021;21(1). http://doi.org/10.1186/s12890-021-01664-7

adjust to

19. Wan J, Zhang Q, Li C, Lin J. Prevalence of and risk factors for asthma among people aged 45 and older in China: a cross-sectional study. Bmc Pulm Med. 2021;21:1-9.

---

## [Editor Report · Decision Letter 2]

11 Sep 2025

Metabolic Syndrome, Social Isolation, and Sarcopenia in Mild Cognitive Impairment: A Multi-Faceted Analysis of Risk Factors and Mediating Pathways

PONE-D-25-33355R2

Dear Dr. Zhongqiang Guo,

We’re pleased to inform you that your manuscript has been judged scientifically suitable for publication and will be formally accepted for publication once it meets all outstanding technical requirements.

Kind regards,

Marwan Salih Al-Nimer, MD, PhD

Academic Editor

PLOS ONE

Additional Editor Comments (optional):

No comments
---

## [Editor Report · Acceptance letter]

PONE-D-25-33355R2

PLOS ONE

Dear Dr. Guo,

I'm pleased to inform you that your manuscript has been deemed suitable for publication in PLOS ONE. Congratulations! Your manuscript is now being handed over to our production team.

Kind regards,

on behalf of

Professor Marwan Salih Al-Nimer

Academic Editor

PLOS ONE